# Reproductive Activity of Socorro Island Merino Ewes and Their Crosses with Pelibuey under Heat Stress Conditions

**DOI:** 10.3390/ani14101405

**Published:** 2024-05-08

**Authors:** Arturo César García-Casillas, Omar Francisco Prado-Rebolledo, María Isabel Carrillo-Díaz, José Luis Zepeda-Batista, Carlos Eduardo Barajas-Saucedo, Juan Augusto Hernández-Rivera

**Affiliations:** 1Faculty of Veterinary Medicine and Animal Husbandry, University of Colima, Tecoman 28930, Colima, Mexico; cesargarciacasillas@hotmail.com (A.C.G.-C.); omarpr@ucol.mx (O.F.P.-R.); mcarrillo13@ucol.mx (M.I.C.-D.); jzepeda15@ucol.mx (J.L.Z.-B.); 2Faculty of Chemical Sciences, University of Colima, Coquimatlan 28400, Colima, Mexico; carlos_barajas24@ucol.mx

**Keywords:** heat stress, Socorro Island Merino ewes, crossbred, progesterone, oestrus

## Abstract

**Simple Summary:**

In order to increase the fertility rate of ewes throughout the year, expensive assisted reproduction programs are used during the spring–summer season despite the hot conditions in the tropics that can negatively affect them. Due to the photoperiod, ewes tend to mate only during the autumn–winter months, when daylight hours are reduced. Merino Socorro Island Merino ewes and their crosses with Pelibuey under warm conditions can significantly help sheep production units located in the tropics of cancer because they show reproductive activity during the spring and autumn seasons in the tropics. In addition, the reproductive efficiency of flocks could be improved by using oestrus synchronization protocols based on timed artificial insemination programs or natural mating.

**Abstract:**

An experiment was carried out to evaluate the effect of spring and autumn seasons on the reproductive activity of Merino Socorro Island ewes and their crosses with Pelibuey under heat stress (**HS**) conditions in the tropics. All ewes (*n* = 80) were randomly assigned to one of two breeds during the first and second periods, respectively: (1) Twenty Socorro Island Merino ewes (**SIM**) and (2) 20 Pelibuey Crossbred ewes (**PBC**). Animals were fed the same diet and given water ad libitum. All statistical analyses were performed using SAS statistical software 9.12 procedures. In both seasons, a mean of more than 80 U of maximum THI was obtained, while in spring and autumn, the minimum THI exceeded 30 and 40 U, respectively. All animals were in oestrus and ovulated in both seasons. The frequency of animals in spring during the first 48 h of oestrus expression was greater (*p* < 0.05) than 48–55 h but similar (*p* > 0.05) than 55–65 h; in autumn during the first 48 h and 48–55 h were similar (*p* > 0.05), but different (*p* < 0.05) than 55–65 h. The duration of oestrus expression was longer in the spring than in the autumn (*p* < 0.05). The frequency of animals was higher (*p* < 0.05) in SIM than in PBC ewes during the first oestrus cycle (1–17 d) and was also higher (*p* < 0.05) in PBC than in SIM ewes during the second oestrus cycle (18–35 d). The SIM ewes produced more (*p* < 0.05) progesterone (**P_4_**) than the PBC ewes. During the sampling days of the oestrus cycle, more P_4_ was created in autumn than in spring (*p* < 0.05). Both breeds showed severe HS. In the future, ewes treated under assisted reproductive programs in the tropics may improve reproductive efficiency.

## 1. Introduction

Heat stress (**HS**) reduces reproductive activity in sheep in desert, semi-desert, and tropical areas worldwide [1]. The sheep under HS present a decrease in fertility, fetal development, and growth, as well as low weight gain and feed efficiency in the fattening stage [2]. Sheep exposure to environmental temperatures higher than 32 °C can decrease fertility rates [3]. A 6-h exposure to HS during ovulation decreases the number of transferable embryos [4]. Some breeds of sheep have shown tolerance to HS; for example, it has been demonstrated that breeds that evolved in warm climates, such as Pelibuey sheep, can thermoregulate their body temperature better than breeds from temperate or cold climates, such as Suffolk sheep [5]. In fact, under thermoneutral conditions, several researchers have reported oestrus rates between 75–89%, oestrus expression between 36–38 h in breeds such as Blackbelly and Pelibuey [6,7], while Pelibuey and Dorper ewes under HS conditions have achieved oestrus rates up to 100%, oestrus expression no later than 48 h, number of oestrus around 5 in 90 d, oestrus cycle length 18 d, and ovulation rate of 100% [1,8].

HS is often described as a temperature–humidity index (**THI**) of 72 or above [2]. In a study performed by Solórzano-Montilla et al. [9] during the spring and autumn seasons in Venezuela, they evaluated the THI, ambient temperature (**AT**), and relative humidity (**RH**) response of 24 West African ewes with ~19 kg live weight. The ewes were placed in paddocks with artificial shade based on awnings with shade netting at 70% light penetration and without shade to take advantage of grazing. These authors collected AT data in both treatments during the morning (09:00 h) and afternoon (13:00 h) until obtaining an average of 27.0 and 33.0 °C, respectively, in the shaded animals. On the other hand, these same authors, Solórzano-Montilla et al. [9], obtained THI averages during the morning and afternoon of 77 and 79 units; likewise, a RH of 76 and 49% was obtained, respectively. In another study conducted by Macías-Cruz et al. [10] with Katahdin and Pelibuey sheep in northwestern Mexico during the summer, considered the hottest months of the year. These animals were in pens provided with steel mesh walls to facilitate airflow and shade made of galvanized sheeting so that some climatic variables such as THI, AT, and RH were evaluated, reporting the highest peaks of the study; in August, a maximum THI of 84 units and a maximum AT of 41.5 °C during the afternoon hours were reported on average.

On the other hand, Sejian et al. [11] reported a decrease in thyroid hormone concentration in Malpura sheep placed in a tropical environment, observing a more evident decrease in thyroxine (**T_4_**) than in triiodothyronine (**T_3_**). T_3_ and T_4_ levels usually increase in low ambient temperatures; the opposite happens in high temperatures [12,13]. Additionally, feed intake, metabolic rate, and productivity decrease [14] because animals under HS conditions suppress the activity of the appetite center located in the hypothalamus, which stimulates the synthesis and release of tyrosine inhibitory factor [2]. Several studies have reported controversial results, although all agree that when animals are subjected to HS conditions, a decrease in thyroid hormone levels is expected [2,9,12,15,16]. Other authors report high levels of thyroid hormones in small ruminants under thermoneutral conditions [12,14,17,18].

On the other hand, Silva-Avila et al. [13], in a study conducted on hair sheep in southern Sonora, evaluated the T_4_ response during the summer (hot environment; 50.0 ng/mL) and autumn (calm environment; 63.6 ng/mL) months, obtained a maximum THI of 84 and 77 U in summer and autumn, respectively, and they mentioned that the environmental conditions observed during the summer were not favorable for the animals. In addition, decreased T_3_ and T_4_ levels can affect reproductive function by interfering with ovarian follicle development, the oestrus cycle, and ovulation [19]. Therefore, it can interfere with the function of the hypothalamic–pituitary–gonadal axis and cause disturbances in the secretion of progesterone (**P_4_**), a steroid hormone produced by the ovaries that is essential for maintaining pregnancy and supporting reproductive processes in sheep [20]. Changes in P_4_ levels due to HS can lead to irregular oestrus cycles, decreased fertility, and reduced reproductive performance in ewes [21]. A study conducted in Nigeria with Yankassa ewes during dry (heat stress conditions) and wet (thermoneutral conditions) weather gave similar results of 7.3 and 6.7 ng/mL of progesterone, respectively, even though water and feed availability was greater during wet weather than dry weather [22]. In contrast, Macías-Cruz et al. [8] reported significant differences in serum P_4_ levels between days 5–14 of the oestrus cycle in Pelibuey ewes subjected to HS in the spring and autumn, reaching a peak in the autumn on day 11 with 8.5 ng/mL of P_4_, almost 3 ng/mL higher than in the spring at the same time.

Socorro Island Merino (**SIM**) sheep were introduced in 1869 and subsequently abandoned by the Australian settlers; gradually, in the absence of zootechnical management, they returned to their feral situation and, under the extreme tropical conditions typical of the island, suffered from lack of fresh water and food [23], this allowed the biotype to fix its characteristics related to adaptation to adverse environmental conditions. Also, the initial flock was small, which caused inbreeding due to prolonged genetic isolation, natural selection, and genetic drift [23]. The Faculty of Veterinary Medicine and Animal Husbandry of the University of Colima has the only SIM sheep population in the world. No studies have evaluated the circannual reproductive activity of SIM ewes under HS conditions. Nevertheless, it has been empirically observed that SIM ewes and their crosses with Pelibuey show continuous oestrus cycles throughout the year, apparently losing the effect of photoperiod. Traditionally, in wool sheep, the length and availability of light have a more significant influence during the autumn and winter months due to their increased reproductive activity [24]. Therefore, the objective of the present study was to evaluate the effect of spring and autumn seasons on the reproductive activity of Merino Socorro Island ewes and their crosses with Pelibuey under HS conditions in the tropics.

## 2. Materials and Methods

### 2.1. Location and Study Area

The study was carried out in the Sheep Station of the Faculty of Veterinary Medicine and Animal Husbandry of the University of Colima between parallels 18°40′ and 19°08′ north longitude, meridians 103°37′ and 103°59′ west, and altitudes between 0 and 1200 m [25]. This area has a tropical climate, with an average temperature of 26 °C and rainfall of 750 mm/year [26].

### 2.2. Treatments and Experimental Animals

The study lasted 70 days and was divided into two periods. The first period was during the spring (9 May to 6 June 2016); the second period corresponded to the autumn (31 October to 29 November 2016). During the first period, forty ewes were assigned to one of two genetic groups: (1) 20 SIM ewes and (2) 20 Pelibuey Crossbred (PBC) ewes, considering animals without foot breeding, clinically healthy, weight (~27.4 ± 0.8 kg) and age (~2 ± 0.1 years old) before the start of the study. In the second period, all available ewes (n = 40) were selected using the same criteria as in the first period. The ewes were housed under the same feeding regime based on chopped king grass (Pennisetum purpureum, CT-115), alfalfa, total mixed ration as a supplement, and water *ad libitum*. The housing pen measured 9 m long × 5.33 m wide, with a height of approximately 1.10 m, and a semi-water shade that measured 2.73 m at the highest area and 1.60 m at the lowest area, covering 50% of the pen with shade. The feeders were 8-inch PVC tubes cut in half to simulate a wooden-bottomed canoe. The water troughs were plastic drums with approximately 80 L of water cut in half.

### 2.3. Reproductive Activity

The ewes were separated from the males before the start of the study; then, to determine the reproductive activity of the ewes, heat detection was performed using three Socorro Island Merino rams as markers with an apron over the chest to avoid copulation. All ewes were exposed twice daily for 30 min throughout the experiment (7:00 and 14:00 h), and a ewe was in oestrus if she showed no movement reflex to mount. They were temporarily separated from the group until the end of the detection period. Afterward, they returned to the group with the other ewes not in oestrus. From this information, the following study variables were calculated for each season: oestrus cycle length in days based on the classification of Sasa et al. [27] (standard [15–19 day], short [≤14 day], long [20–26 day] and multiple [≥27 day]). The duration of oestrus expression was based on the modified classification of Rodrigues et al. [19] (standard [24–36 h], short [<24 h] and long [>36 h]).

### 2.4. Hormone Analysis

Blood samples were collected at each season to determine P_4_, T_3_, and T_4_ concentrations by repeated measurements over time to assess ovulation and metabolism. The procedure was performed twice a week in the morning before feeding. Physical restraint of the ewes and blood was collected by jugular venipuncture using needles and 6 mL Vacutainer^®^ tubes without anticoagulant. The blood samples were centrifuged (Megafuge 2.0R, Heraeus, Hanau, Germany) at 3500× *g* for 15 min at five °C to obtain the serum collected in previously labeled 1.5 mL vials. Serum samples were then stored at −20 °C and used for the determination of P_4_, T_4_, and T_3_ by a method using validated commercial kits (Monobind Inc., Lake Forest, CA, USA) in an automated ELISA analyzer (Multiskan™ FC, Thermo Fisher Scientific, Waltham, MA, USA). The kit for T_3_ had a sensitivity of 0.04 ng/mL, an intra-assay CV of 5.4%, and an inter-assay CV of 6.7%, while the kit for T_4_ had a sensitivity of 0.1 ng/mL, an intra-assay CV of 1.6% and an inter-assay CV of 6.1%. Lastly, the kit for P_4_ had a sensitivity of 0.105 ng/mL, an intra-assay CV of 3.8%, and an inter-assay CV of 7.5%. Finally, an ewe was considered to have ovulated if the P_4_ concentration was ≥1.0 ng/mL in at least two consecutive samples after presenting P_4_ levels < 1.0 ng/mL [28]. With this information, the following variables were generated by season: percentage of ovulating ewes (at least one ovulation per season) and number of oestrus. 

Corpus luteum activity was evaluated by measuring serum P_4_ levels during one oestrus cycle each season. In spring and autumn, the onset of oestrus was detected in the ewes, and two days later, another blood sample was taken from the jugular vein; this activity was repeated every three days until the onset of oestrus was detected again (days 2, 5, 8, 11, 14, and 17 of the cycle). The equipment, extraction procedure, and sample handling used to obtain serum and assess P_4_ levels were comparable to those previously reported for ovulation and metabolism.

### 2.5. Environmental Responses

The environmental responses such as AT (°C) and RH (%) were recorded every 15 min, and these data were obtained from the meteorological station of the Faculty of Biological and Agricultural Sciences of the University of Colima, located on the Tecoman Campus. With the variables obtained, the temperature-humidity index was calculated as an indicator of HS with the following formula [29]:*THI* = 0.81 *AT* + *RH* (*AT* − 14.4) + 46.4
where

*THI* = temperature-humidity index, U;

*AT* = ambient temperature, °C;

*RH* = relative humidity, %.

### 2.6. Statistical Analysis

All statistical analyses will be carried out using the procedures of the SAS statistical program [30]. Fisher’s exact test was used to analyze variables expressed as percentages, such as reproductive activity, using PROC FREQ. For continuous variables about hormonal variables, the UNIVARIATE procedure was used to demonstrate their normal distribution. A completely randomized design with repeated measurements over time was then used and analyzed with PROC MIXED. The statistical model was as follows:*Y_ijk_* = *μ* + *B_i_* + *S_j_* + *RMT_k_* + *Ɛ_ijk_*
where

*Y_ijk_* = hormone concentration;

*Μ* = general mean;

*B_i_* = fixed breed effect;

*S_j_* = fixed seasonal effect;

*BMT_k_* = effect of repeated measurement in time;

*Ɛ_ijk_* = experimental error.

Least squares mean and standard errors of means are reported, and significance is accepted if *p* < 0.05, with a trend accepted if 0.05 < *p* < 0.10.

## 3. Results

### 3.1. Environmental Responses

Figure 1 shows the weekly maximum and minimum values of the THI, AT, and RH during spring (above) and autumn (below). The average maximum and minimum AT obtained during the five weeks of the spring study were 39.7 and 23.0 °C, respectively, with a maximum AT of 40 °C in the second and third week of the study and a minimum AT of 24.5 °C in the fifth week of the study. In the case of RH, an average of 89% and 33% was obtained, respectively, and a maximum RH peak of 90% in week three of the study; additionally, a minimum HR of 34.4% was observed in that same week. On the other hand, the average obtained maximum and minimum THI during the experimental time were 87 and 73 units, respectively, with a maximum THI of 89.4 units in week three and a minimum THI of 75.1 units in week five. The average maximum and minimum AT during the study weeks in autumn was 34 and 24 °C, respectively, while RH was 97% and 49%, respectively. On the other hand, the average maximum and minimum THI during the experimental time were 84 and 74 units, respectively.

### 3.2. Reproductive Activity

Figure 2 shows the frequency of oestrus rate by season. One hundred percent of animals that entered oestrus at least once during the study period were included.

Table 1 shows the frequency of animals in terms of the duration of oestrus expression, the number of oestrus, and the duration of the oestrus cycle at breed and in two seasons. No differences (*p* > 0.05) were observed between breeds concerning the time recorded during oestrus expression, while during spring, 48.5% of the animals expressed oestrus in less than 48 h, similar (*p* > 0.05) to the group >55 < 65 h with 33.3% of the animals, which were higher (*p* < 0.05) than the group >48 < 55 h with 18.2% of the animals. On the other hand, similar results (*p* > 0.05) showed the groups <48 and >48 < 55 h with 45.9% and 40.5% of animals, respectively, but were higher (*p* < 0.05) than the group >55 < 65 h with 13.5% of animals. The frequency of animals on the number of oestrus obtained during the study was similar between breeds and seasons (*p* > 0.05). The frequency of animals was higher (*p* < 0.05) in SIM than in PBC during the first oestrus cycle (1–17 d), while during the second oestrus cycle (18–35 d), PBC was higher (*p* < 0.05) than SIM. Likewise, no differences were recorded between seasons (*p* > 0.05).

In Table 2, no differences (*p* > 0.05) were observed between the number of oestrus on breed and season, also no differences between the days of the oestrus cycle on breed, but on season, the autumn tended to be higher (*p* = 0.0835) than spring season. SIM ewes tended to increase the hours of oestrus expression (*p* = 0.0876) than in the group of PBC ewes. Finally, the duration of oestrus expression in spring is higher (*p* < 0.05) than in autumn.

### 3.3. Hormone Analysis

The results of the hormonal response are presented in Table 3. No differences (*p* > 0.05) were observed between breeds in the concentration of T_3_ and T_4_. SIM ewes showed higher (*p* < 0.05) P_4_ than PBC, while no differences (*p* > 0.05) were observed between seasons. T_3_ was higher (*p* < 0.05) in spring than in autumn, with 1.55 and 1.24 ng/dL, respectively. On the other hand, in contrast, the obtained T_4_ concentration was lower (*p* < 0.05) in spring, with 4.95 ng/mL, compared to 5.48 ng/mL in autumn. 

In Figure 3, the P_4_ concentrations were affected (*p* < 0.05) by season and breed effects in the days of the oestrus cycle. Between days 5 and 14 of the cycle, differences (*p* < 0.05) in P_4_ concentration were observed between seasons and breeds. On days 8 to 14, the P_4_ concentration was higher (*p* < 0.05) in SIM ewes than in PBC ewes and was also higher (*p* < 0.05) in autumn ewes compared to spring ewes.

## 4. Discussion

### 4.1. Environmental Responses

According to the evidence in Figure 1, the maximum THI was above 80 units in both spring and autumn. HS has a detrimental effect on the reproductive processes of sheep. As the temperature rises, the animal cools by sweating and panting [15]. HS inhibits reproductive function by reducing feed intake and increasing water consumption. These two variables can affect metabolic processes and negatively affect reproductive processes in the flock [31]. Avendaño-Reyes et al. [32] describe that HS-free animals have a THI < 72 units, 73–78 is moderate HS, 78–84 is severe HS, and >84 is critical to cause death. Therefore, our SIM and PBC ewes were under severe HS, i.e., both breeds had no respite from hostile weather conditions during the study. Similarly, in northern Mexico, Macías-Cruz et al. [10] reported 84 THI units in hair sheep during the hottest hours of the day. In these situations, the adverse effects of HS on reproductive activity are well known [33,34,35,36,37]. Still, environmental conditions such as direct or indirect solar radiation and wind speed can also exacerbate such effects. The increased susceptibility of sheep to HS is becoming a significant concern as temperatures rise due to fossil fuel-driven climate change. Then, although the thermoneutrality of sheep is between 12–17 °C [38], the reality in our study is that the maximum temperature in the spring reached almost 40 °C. At the same time, in autumn, it exceeded 30 °C, and both cases were well above average. According to this, rising temperatures have resulted in more severe and prolonged HS in our ewes. The temperatures are increasing yearly, which is predicted to continue [39]. The fertility rate may decrease by 3% at temperatures above 32 °C, which could be influenced by failed fertilization or embryo survival [40]. Likewise, the normal RH range is between 30–70% [41], but the maximum RH reached almost 90% in spring and exceeded in autumn. 

### 4.2. Reproductive Activity

The reproductive activity showed an oestrus rate of 100% in both seasons and animal breeds, which is consistent with what has been reported by some authors in the same species and hostile environment [8,42]. At least, it has been observed that during the summer, HS does not affect the presence of corpora lutea or oestrus rate in Pelibuey and Suffolk ewes [43]. Interestingly, all results do not seem to suffer any adverse effects from HS, which coincides with the report by De la Isla Herrera et al. [44], which mentioned that high temperatures are not an environmental factor that conditions oestrus activity in Pelibuey ewes. Therefore, the percentage mentioned above in our sheep completes the previous empirical discoveries by showing many potentials for the manipulation of the oestrus cycle, multiple ovulation, and embryo transfer, for the preservation of breeds, and for the increase in productive and reproductive efficiency during spring and autumn in tropical conditions. As a result, the implementation of assisted reproduction programs seems to be an urgent task, given the various situations that cause a discharge in the number of animals, such as specimen sales, deaths due to nutrimental or reproductive management, diseases or accidents, genetic problems, or HS conditions. That is, optimizing management to improve the productive and reproductive potential of SIM ewes may be essential for producers, especially if they want to maintain or enhance the reproductive parameters of the flock. Therefore, the SIM ewes may have adapted over time in a similar way to the Pelibuey sheep, where they are characterized by being continuous poly-oestrus during the natural reproductive season (summer-autumn) [10,45], and then remembering that the living conditions on this island were hostile. Therefore, although SIM and PBC ewes showed reproductive activity during the spring and autumn months, eliminating the effect of photoperiod, it is essential to complement the circannual reproductive activity by evaluating the winter and summer seasons. During the spring, the frequency of animals on the duration of oestrus expression was high in less than 48 h and 55–65 h.

In contrast, in autumn, the frequency of animals on the same variable was high at less than 48 h and more than 48–55 h. Likewise, the duration of oestrus expression tended to be almost five hours higher in SIM ewes than in PBC ewes. This difference could be related to the SIM ewes’ greater metabolic capacity and the survival factors they developed on the island [23]. Then, the duration of oestrus expression was almost seven hours higher in spring than in autumn. This difference is due to HS, which causes several problems in female oestrus activity, including oestrus expression, delayed onset, and oestrus signals for short periods [46]. In addition, the effects of HS may affect follicular development, follicular dominance, and ovulation in these species [47]. In both cases of breed and season factors, the duration of oestrus expression was long according to the classification of Rodrigues et al. [19]. The amplitude of time in the duration of oestrus expression offers excellent possibilities for increasing the fertility rate in SIM ewes, especially during insemination and mating in tropical conditions.

On the other hand, the frequency of SIM ewes during the first period (1–17 d) of the oestrus cycle was more than PBC ewes. Then, PBC ewes during the second period (18–35 d) of the oestrus cycle were more than SIM ewes. According to the classification of Sasa et al. [27], the timing of both oestrus cycles determined during the study period was standard, in agreement with the average of the oestrus cycle tended to be higher in the autumn than in the spring. In a coincidence, Macías-Cruz et al. [48] reported a similar average to our study in 17 d of the oestrus cycle in Pelibuey ewes under HS in northern Mexico. 

### 4.3. Hormone Analysis

Thyroid hormones are chemical messengers widely related to the metabolism of animals. Changes in T_3_ and T_4_ concentrations were not observed between breeds because both were managed and housed under the same HS conditions. However, T_3_ levels in our study were 25% higher in the spring than in the autumn months. In contrast, T_4_ levels were 11% higher in the autumn months than in the spring months. This behavior is expected because T_4_ is the primary hormone secreted by the thyroid gland and converted to T_3_, the active form of thyroid hormone. Therefore, T_3_ is more sensitive than T_4_, especially at high temperatures [49]. In this scenario, higher serum T_3_ concentrations usually occur in HS, as was the case in spring, the hottest and most hostile season in our study, where, interestingly, serum T_4_ levels were low.

In contrast, when ITH units were lower in the autumn, T_4_ levels were higher than T_3_. Our study did not measure food consumption, but it is well-known that T_3_ directly stimulates feed intake at the hypothalamic level. In contrast, the quantity and quality of the feed consumed are essential factors influencing thyroid hormone concentrations [14]. Also, T_4_ increases voluntary feed intake in ewes [50]. In contrast, during the presence of HS, the levels of T_3_ and T_4_ usually decrease. This means that as thyroid hormones decrease in HS, appetite is reduced by the hypothalamic satiety center, ruminal movements, feed passage rate, and dry matter digestibility decrease, leading to reproductive problems [51,52,53,54]. In addition, thyroid hormones influence ovarian steroidogenesis, ovulation, and corpus luteum function. It also promotes GnRH release to regulate ewes’ hypothalamic-pituitary-gonadal axis and cyclicity. Finally, more reports on reference values for T_3_ and T_4_ must be made in Mexico, especially in small ruminants under HS conditions.

In general, the behavior of P_4_ in ruminants is not usually as sensitive to the effects of HS as thyroid hormones. Thus, in our study, no apparent changes in P_4_ concentration were observed during spring and autumn. On the contrary, P_4_ in serum was consistently higher in SIM ewes than in PBC ewes by 32%. Despite this, our P_4_ levels are below those observed in the literature [8,22] due to severe HS. From a biochemical point of view, cholesterol acts as a precursor for P_4_ synthesis [55], which indicates the amount of feed consumed, i.e., higher feed consumption can increase serum cholesterol levels and then P_4_. However, as mentioned above, feed consumption is reduced under HS, and a generalized decrease in P_4_ synthesis could occur in SIM and PBC ewes under tropical conditions. According to the P_4_ synthesis obtained during the repeated measurements in our study, 100% of the SIM and PBC ewes showed an ovulation rate at least once during the study period. In Pelibuey ewes, 38.4% higher ovulation rates were reported in the summer and autumn months compared to the winter and spring months [44]. A decrease in P_4_ during seasonal anoestrus may indicate the effect of photoperiod or the absence of cyclicity. Conversely, an increase in P_4_ may induce a continuous cycle in ewes by negative feedback in the hypothalamus until pregnancy is achieved or maintained by mechanisms related to pregnancy recognition via interferon-τ or until luteolysis is induced via prostaglandins F2α. If so, folliculogenesis culminates in the formation of a Graaf follicle until ovulation is reached, but not before estrogen receptors at the level of the theca cells in the ovary are increased to raise serum estradiol levels, induce a positive feedback effect in the hypothalamus for GnRH release, or simply manifest signs of oestrus [56,57,58]. 

The assessment of corpus luteum activity is entirely unprecedented. Marked differences between SIM and PBC ewes and between spring and autumn seasons according to the sample days of the oestrus cycle agree with authors from northern Mexico [8]. Although SIM and PBC ewes showed clear signs of oestrus and ovulation at least once during the study period, they likely show some reproductive seasonality according to P_4_ concentration, which was lower in PBC ewes or in the spring season. This is due to the pronounced seasonality that wool sheep have inherited from their ancestors, with the reproductive season traditionally occurring mainly in the autumn and winter months and, to a lesser extent or not at all, in the spring and summer months. Consistent with our study, a study conducted on Yankassa sheep in Nigeria showed higher serum P_4_ during dry months and lower serum P_4_ during wet months [21]. The same authors mentioned that P_4_ typically prepares the uterus for the zygote’s arrival from the oviduct’s ampulla, its implantation, and the subsequent embryonic and fetal development during pregnancy. Finally, given that HS conditions exist almost year-round, given the decreases in P_4_ found in this study based on the literature, and ignoring the possibility that these decreases are specific to our breeds, P_4_ can play an essential role in assisted reproduction programs in most oestrus synchronization protocols. To aid in oestrus synchronization, it can be administered by a variety of routes, including the controlled internal drug release (CIDR) device, medroxyprogesterone acetate (MAP), and fluorogestone acetate (FGA) [59]. In addition, it is common practice to supplement P_4_-based protocols with gonadotropins such as eCG, pregnant mare serum gonadotropin (PMSG), and gonadotropin-releasing hormone (GnRH) at the end of the synchronization phase to enhance fertility [59,60,61].

## 5. Conclusions

The Socorro Island Merino ewes and their crosses with Pelibuey experienced severe heat stress. All animals ovulated and showed signs of oestrus at least once during the study; therefore, heat stress or photoperiod had no apparent effect on the reproductive activity of the ewes. In both study seasons, the distribution of animals is exceptionally high during the first 48 h of oestrus expression. The Socorro Island Merino ewes produced higher amounts of progesterone than Pelibuey Crossbred ewes; this may help maintain early gestation in heat-stressed animals. However, more progesterone was made during the oestrus cycle days in the autumn than in the spring. In the future, the use of assisted reproduction programs such as laparoscopically timed artificial insemination-based oestrus synchronization protocols or the use of multiple ovulation and embryo transfer protocols could improve the reproductive efficiency of ewes.

## Figures and Tables

**Figure 1 animals-14-01405-f001:**
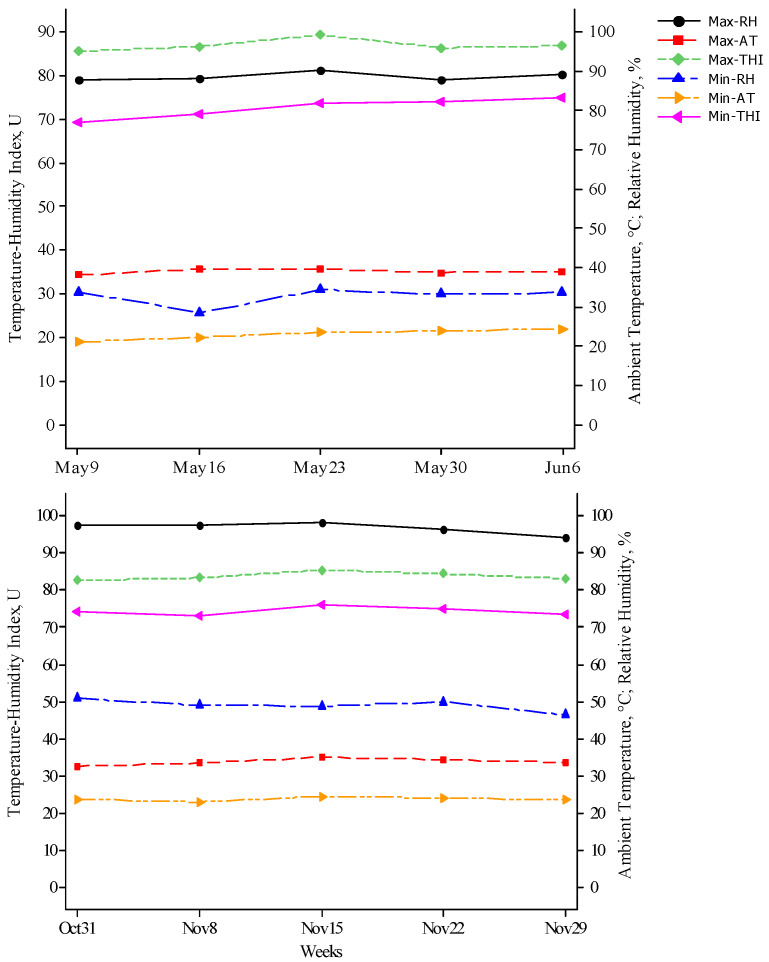
Weekly maximum and minimum values of the temperature–humidity index (THI), ambient temperature (AT), and relative humidity (RH) during spring (**above**) and autumn (**below**).

**Figure 2 animals-14-01405-f002:**
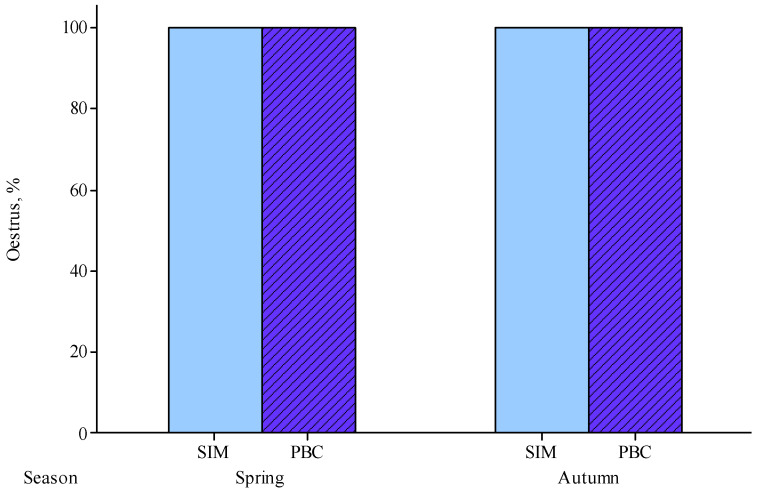
Oestrus rate by season and breed. SIM = Socorro Island Merino ewes; PBC = Pelibuey Crossbred ewes.

**Figure 3 animals-14-01405-f003:**
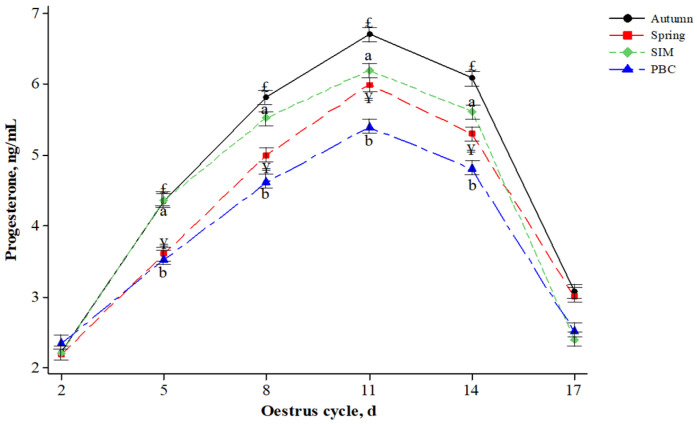
Effect of seasons and breed on progesterone concentration on oestrus cycle days in Socorro Island Merino (SIM) ewes and their crosses with Pelibuey (PBC). ^a,b,£,¥^ Literals and symbols indicate differences between breeds and seasons, respectively, within each day of the oestrus cycle at *p* < 0.05.

**Table 1 animals-14-01405-t001:** Oestrus expression, number of oestrus, and oestrus cycle length among breeds and seasons.

	Breed, %		Season, %	
	SIM	PBC	Spring	Autumn
Ewes, *n*	40	40	40	40
Oestrus expression, h				
<48	37.5	60.0	48.5 ^c^	45.9 ^c^
>48 < 55	35.0	23.3	18.2 ^d^	40.5 ^c^
>55 < 65	27.5	16.7	33.3 ^c^	13.5 ^d^
Oestrus, *n*				
1	52.5	60.0	60.6	51.4
2	37.5	36.7	33.3	40.5
3	10.0	3.3	6.1	8.1
Oestrus cycle, d				
1–17	42.5 ^a^	23.3 ^b^	80.0	60.0
18–35	57.5 ^b^	76.7 ^a^	20.0	40.0

^a,b^ Different literals within the row on breed, as well as on season by oestrus (*n*) and oestrus cycle (d), indicate a difference (*p* < 0.05). ^c,d^ Different literals within the same column on the season by oestrus expression (h) indicate a difference (*p* < 0.05). SIM = Socorro Island Merino ewes; PBC = Pelibuey Crossbred ewes.

**Table 2 animals-14-01405-t002:** Effects of breed and season on oestrus activity.

	Breed				Season			
	SIM	PBC	SE	*p*-Value	Spring	Autumn	SE	*p*-Value
Oestrus, *n*	1.89	1.68	0.2083	0.3071	1.65	1.93	0.2078	0.1857
Oestrus cycle, d	16.6	16.7	0.4009	0.8168	16.3	17.0	0.3975	0.0835
Oestrus expression, h	55.8	51.1	2.7332	0.0876	56.8	50.1	2.7096	0.0152

SIM = Socorro Island Merino ewes; PBC = Pelibuey Crossbred ewes.

**Table 3 animals-14-01405-t003:** Effects of breeds and season on hormone responses.

	Breeds				Season			
	SIM	PBC	SE	*p*-Value	Spring	Autumn	SE	*p*-Value
Progesterone, ng/mL	5.54	4.20	0.609	0.0301	4.67	5.08	0.609	0.4801
Triiodothyronine, ng/mL	1.41	1.37	0.048	0.4348	1.55	1.24	0.048	<0.0001
Thyroxine, ng/mL	5.22	5.20	0.079	0.8132	4.95	5.48	0.079	<0.0001

SIM = Socorro Island Merino ewes; PBC = Pelibuey Crossbred ewes.

## Data Availability

Data are contained within the article.

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
