# Peer review of "Reproductive Activity of Socorro Island Merino Ewes and Their Crosses with Pelibuey under Heat Stress Conditions"

_animals, 2024, doi:10.3390/ani14101405_

Round 1

Reviewer 1 Report

Comments and Suggestions for Authors

38 Introduction It seems necessary to introduce the photoperiod effect in the small ruminants and its possible impact on reproduction according to the location of your experience. Morover, it could be necessary to reminder the interest of thyroid hormones evaluation to describe the reproductive activity. You mention changes but you need to described better the effect of heat stress on hormonal concentrations.

62/74 : could you give some more informations about the zootechnical/reproduction  performances of the two used breeds (Socorro Island Merino and Pelibuey) : meat production, usual periods of lambing …

81 attending you hace consider two study period (spring and autumn) it could be better to descrisbed the average rainfall and temperatures durin the two periods. Could you also etter define the calendar limits of spring and autumn.

87 you need to define the dates of experiment according to the periods of spring and autumn

88 could you give some informations about the body condition score of the ewes ?

102 three males by group of 20 ewes ?

103 could you tell if the females were separated from the male before the periods of study

106 that means the ewe go back to the group after 30 minutes ?

109 time or duration ?

164 could you tell if the presented values are hourly,  daily or weekly values ?

179 according to the figure, it seems that the values are recorded once a week : could you make a graph with the daily value ? It seems better to present the means of the different parameters.

188 it’s 48.5 and not 48 % : please adjust your foiguures according to the values presented in the table 1

200 attending the absence of differences, such figure is not necessary.

202 it surprising that there is no significant differences of the lenght of estrus cycle  during spring (80 vs 20). It’s also true for the estrus expression for PBI ewes.

209 and 219 E.E. : what does it means ?

216 The following sentence is not understandable : But if 5 ng/mL were recorded during the 216 spring in comparison (P

<0.05) with autumn where 5.48 ng/mL were obtained. S>

219 thyroxine and not thyrosine

234 could you summarized your environmental observations and the comparisons with some other studies. It’s difficult to understand the true differences between the two periods. Nevertheless, such differences are importnat ton understand the differences of reproductive activities and hormonal concentrations.

236 units means day ? again such word is not clear (see my remark line 164)

295 the concentration in autumn months were …

343 why are you speaking on photoperiod and not heat stress ????

348 could you give some specific exemples of assisted breeding conditions ?

Author Response

To whom it may concern;

I am writing to whom it may concern to comment that the authors are grateful in advance for the suggestions made to the manuscript, and in the hope of receiving an approving vote, we bid you a kind farewell.

In the attached file you will find the response sheet to the suggestions, Turnitin analysis to obtain the percentage of plagiarism and the restructured document.

Let me know if I can be of any further assistance.

Sincerely yours,

Juan Augusto Hernández Rivera

Reviewer 2 Report

Comments and Suggestions for Authors

This study describes the reproductive activity of two breeds at two seasons in a tropical area where one breed was introduced many years ago. Each of the observations reported in the manuscript has been reviewed and should be considered. There are several aspects that need substantial correction.

It should focus on the aim and content of the introduction and start immediately with the reproductive description of the sheep in the scenarios in which the study was carried out. Some changes are also needed in the M&M and in the results, as highlighted in the manuscript.

For the discussion, I think this is the weakest point as well as the introduction, as it repeatedly describes the results of other experiments, which, according to the description of the experiment, fit very well into the introduction.

For example, it talks about heat stress, but this is not described in the title and aim of the study. Surprisingly, however, it is described in the first part of the discussion. This should be corrected.

The English language needs to be corrected.

Comments on the Quality of English Language

Should be reviewed and corrected by an English expert

Author Response

(The authors gave the same response as above.)

Round 2

Reviewer 2 Report

Comments and Suggestions for Authors

The comments marked in the first manuscript have been made and the article is therefore accepted.

Comments on the Quality of English Language

no comments